# Transfer Learning Based on Clustering Difference for Dynamic Multi-Objective Optimization

Fangpei Yao and Gai-Ge Wang *

School of Computer Science and Technology, Ocean University of China, Qingdao 266100, China
* Correspondence: wgg@ouc.edu.cn

**Abstract:** Dynamic multi-objective optimization problems (DMOPs) have become a research hotspot in engineering optimization, because their objective functions, constraints, or parameters may change over time, while quickly and accurately tracking the changing Pareto optimal set (POS) during the optimization process. Therefore, solving dynamic multi-objective optimization problems presents great challenges. In recent years, transfer learning has been proved to be one of the effective means to solve dynamic multi-objective optimization problems. However, this paper proposes a new transfer learning method based on clustering difference to solve DMOPs (TCD-DMOEA). Different from the existing methods, it uses the clustering difference strategy to optimize the population quality and reduce the data difference between the target domain and the source domain. On this basis, transfer learning technology is used to accelerate the construction of initialization population. The advantage of the TCD-DMOEA method is that it reduces the possibility of negative transfer and improves the performance of the algorithm by improving the similarity between the source domain and the target domain. Experimental results show that compared with several advanced dynamic multi-objective optimization algorithms based on different benchmark problems, the proposed TCD-DMOEA method can significantly improve the quality of the solution and the convergence speed.

**Keywords:** dynamic multi-objective optimization; evolutionary algorithm; prediction; transfer learning

## 1. Introduction

Dynamic multi-objective optimization problems (DMOPs) [1,2] are optimization problems in which the objective function and decision variables are related to time (environment). Their optimal solution is a set of Pareto optimal solutions that change dynamically with time (environment). Different from solving the static multi-objective optimization problem, when dealing with this kind of optimization problem, it is necessary to not only optimize several conflicting objectives, but also deal with the changes of objective function and constraints at the same time. DMOPs are widely used to solve many real-world problems, and common application domains include scheduling [3,4], control [5], chemistry [6], industry [7] and energy design [8].

An ideal dynamic multi-objective optimization algorithm should contain three necessary parts, namely change detection, change response and multi-objective optimization algorithm (MOEA) [9–12]. When the time variable changes, the algorithm needs to detect the change of the objective function in time and respond to the change according to different types of changes, deal with the optimization problem after the change, and use the multi-objective optimization algorithm to iterate the population, then quickly find DPF and DPS at the current moment. In fact, DMOPs can be regarded as a static multi-objective optimization problem (SMOEA) under a set of discrete time variables that can convert complex dynamic characteristics into static processing, which is more convenient and easier to handle, but its disadvantages are also obvious. That is, the processing speed is slow, and timeliness cannot be guaranteed. When encountering an environment with a high frequency of changes, it often fails to achieve the purpose of quickly tracking the Pareto front,

resulting in poor performance of the algorithm. Therefore, for ideal DMOEAs, a suitable change response mechanism is the top priority in dealing with DMOPs. The existing environmental response strategies can be divided into the following four categories: diversity maintenance, memory strategy, prediction mechanism and transfer learning [13,14].

Diversity maintenance improves algorithm performance mainly by maintaining or improving population diversity. The specific process is shown in Figure 1a. Xu et al. [15] used the perturbation method to divide the decision variables into time-dependent and time-independent according to the dependence of the decision variables on the time parameters, and respectively adopted the optimal solution of the corresponding decision variables of the co-evolution of two subpopulations. In addition, Zhang et al. [16] maintained population diversity by simulating magnetic particles, and then quickly converged to the Pareto front in the current environment. Generally speaking, the diversity maintenance strategy directly adopts the Pareto optimal solution set of the optimization problem in the previous environment as the initial population in the current environment. Liu et al. [17] used an additional auxiliary strategy to maintain diversity that maintains two archives focusing on convergence and diversity, respectively. In addition, for some problems with biased characteristics, an interval mapping strategy is designed to make their solutions have good diversity. Based on this, Liang et al. [18] divided the decision variables into three parts and adopted the methods of diversity maintenance, prediction and diversity introduction, respectively, to generate high-quality offspring individuals and speed up the convergence of the population. Diversity preservation methods have better performance for DMOPs with weaker changes, but when the optimal solution of the historical environment deviates far from the real Pareto front in the current environment, it will lead to poor problem tracking performance.

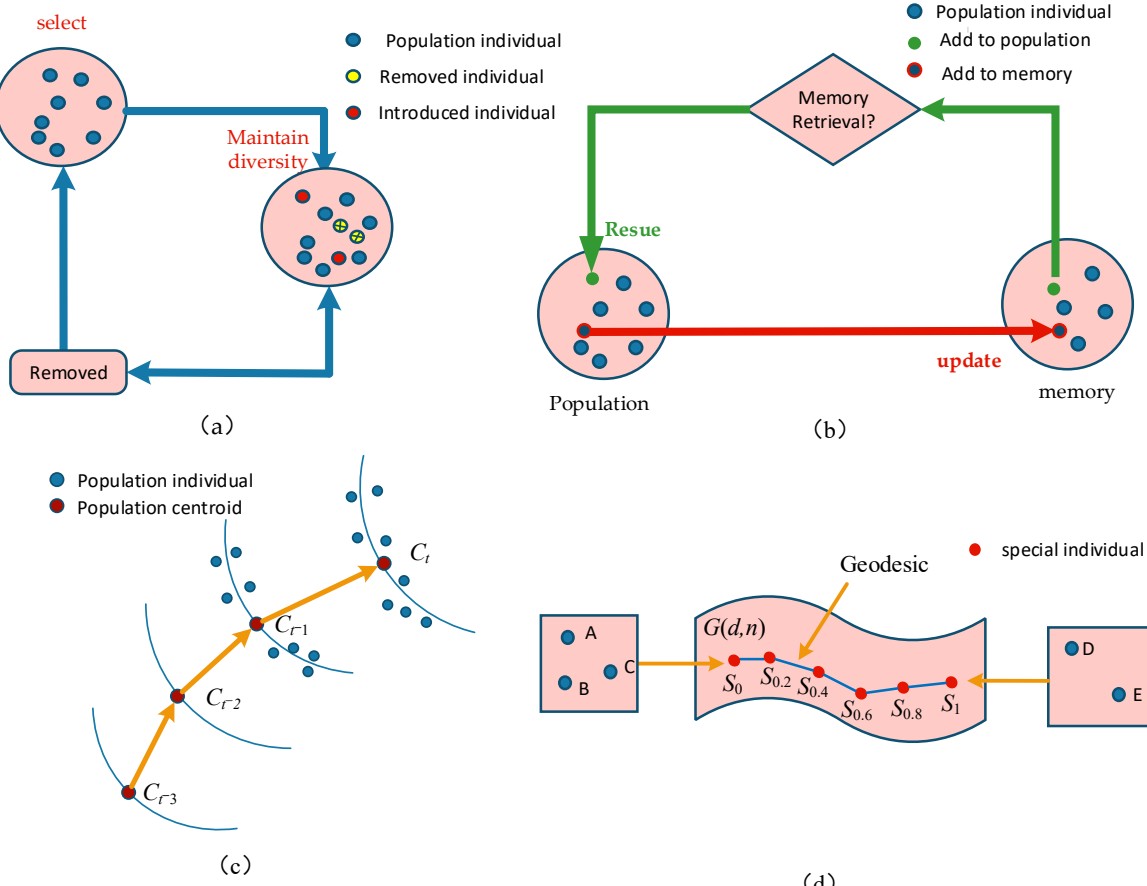

**Figure 1.** Description of environmental response strategies and methods. (**a**) diversity maintenance. (**b**) Memory based approach. (**c**) Prediction based approach (**d**) Manifold transfer learning method.

Methods based on memory strategies and predictive mechanisms are common approaches to address DMOPs; the process is shown in Figure 1b. Chen et al. [19] proposed an evolutionary algorithm for dealing with time-varying constraints and objective functions. The algorithm employs new mating selection and environment selection operators that allow the population to contain both feasible and infeasible solutions and reuse previous solutions based on information obtained from new environments. Yu et al. [20] used polynomial regression predictors to extract linear or non-linear relationships in historical changes to generate good initial populations for new environments. Zou et al. [21] developed a reinforcement learning method to respond to environmental changes according to the severity of the change, which is considered as three states (mild, moderate and severe). The method adopts three actions of knee-based prediction, center-based prediction and local search, and selects a series of actions according to a given state to reorient the population to a new Pareto front (PF).

Based on the evolution information of the optimization problem in the historical environment, the prediction strategy predicts the fitness terrain or the dominant evolution direction of the current environment and provides useful guidance for the evolutionary optimization process, thereby improving the performance of the algorithm. The process of the forecasting model is shown in Figure 1c. Geng et al. [22] designed a group prediction strategy by converting the individual positions at different moments in the same direction of convergence in the target space into time series, predicting the position at the next moment, improving the diversity and effectiveness of the predicted population, and effectively reducing the convergence time of the algorithm after changing the problem. Wang et al. [23] proposed a prediction strategy based on ensemble learning. The strategy has three forecasting models, including linear and non-linear. Given this, Rong et al. [24] constructed a multi model prediction method for the characteristic change types of translation, rotation and composite problems. Compared with the existing three prediction methods, the proposed prediction mechanism has significant advantages in solving most DMOPs. Zheng et al. [25] used different prediction strategies for different decision variables to generate a new initial population according to the different effects of decision variables on convergence and distribution. Ma et al. [26] recently proposed a feature information prediction algorithm for DMOPs. Among them, the joint distribution adaptation model is used to identify the distribution of solutions after environmental changes and create a population in a new environment on this basis.

Although the prediction model is more widely applicable, it inevitably has prediction errors, which affect the accurate guidance of the optimization process. From a statistical point of view, the solution set used to construct the prediction model and the solution set predicted by the prediction model must obey the independent and identical distribution assumption, thus ignoring the non-independence and identical distribution of the data. In view of this, Jiang et al. [27] introduced the transfer learning strategy as the environment response strategy.

Transfer learning makes full use of the problem information with similar characteristics to guide the prediction or classification of current problems and improve recognition accuracy. Jiang et al. [28] proposed a multi-objective dynamic learning method based on evolutionary learning. Using the transfer principal component analysis method, we learn the Pareto optimal solution set in the adjacent historical environment and generate a set of initial populations through the transfer model in the current environment. Experimental results show that this method can accelerate population convergence and accurately track the optimal solution in the new environment. Jiang et al. [29] proposed a fast dynamic multi-objective evolutionary algorithm based on manifold learning. This method combines the memory mechanism with the learning characteristics of manifold transfer to predict the optimal individual in new instances in the process of evolution. The process is shown in Figure 1d. Liu et al. [30] recently proposed the combination of PPS method and transfer learning to improve population prediction. Jiang et al. [31] proposed a method based on individual transfer learning to solve DMOPs. Unlike the existing method, this method

uses a pre-search strategy to filter out some high-quality individuals with good diversity, avoiding negative migration caused by individual agglomeration. Fan et al. [32] also applied transfer learning to solve DMOPs with a large amount of computing, and used alternative auxiliary evolution algorithms, especially MOEA/DEGO, as the baseline to evolve and optimize under limited functional assessment. In addition, transfer learning is used to map the previously archived training data to the current environment to quickly start the proxy model building process, so that the dynamic multi-objective evolutionary algorithm can better adapt to the new environment.

Combining the above transfer learning strategies, the transfer learning strategy has good performance in solving dynamic multi-objective optimization test problems that contain periodic changes and have a large degree of change. However, the application of transfer learning to DMOP solving is still in its infancy. Multiple studies have shown that hybrid change response methods generally perform better than single methods because hybrid methods can handle more diverse dynamic features than single methods. This is demonstrated by the increasing use of mixed strategies in recent work [33].

However, the existing transfer learning methods often need a long training time, which is the main obstacle of some DMOPs. One of the reasons for the slow running speed is that the existing transfer learning methods often realize knowledge reuse by searching the potential space, which will require more parameter settings and consume more computing resources, resulting in a large amount of computing resources wasted on searching low-quality individuals, which greatly increases the possibility of negative transfer [34,35]. If the knowledge possessed by these high-quality individuals can be transferred (from the perspective of convergence and diversity), then more effective and accurate prediction models can be built for the application of DMOPs in various real complex environments.

At the same time, existing dynamic multi-objective optimization algorithms based on forecasting strategies usually use a single forecasting strategy. On the one hand, a single prediction strategy cannot quickly and effectively respond to complex environmental changes; on the other hand, the group diversity generated by a single prediction strategy is poor, and it cannot quickly and effectively track the Pareto front, resulting in the algorithm not being able to quickly converge. Based on the above analysis, in order to reduce the occurrence of negative transfer and improve the running speed, this paper proposes a new environment response mechanism that combines the cluster difference strategy and the transfer learning strategy.

In this paper, the similarity between the source domain and the target domain is improved by adding a clustering difference strategy to predict individuals before transfer learning, thereby reducing the possibility of negative transfer. Then, the transfer learning method TradaBoost [36] is used to build the prediction model. A higher quality initial population is generated through this model, followed by subsequent multi-objective optimization. Therefore, this method is suitable for any population-based multi-objective optimization algorithm and can achieve a large performance improvement. Experiments on different test functions show that the proposed strategy is highly competitive in dealing with problems with different dynamic characteristics and achieves better convergence and distribution.

The contributions of this paper are summarized as follows:

(1) The Pareto solution set at the next moment is predicted by the clustering difference strategy, so as to narrow the difference between the source domain and the target domain of transfer learning, thereby reducing the possibility of negative transfer. Therefore, the preprocessing process of the target domain is very necessary and can make the subsequent transfer learning more efficient.

(2) After the target domain is preprocessed, a sample classifier based on the TradaBoost algorithm is used to extract high-quality populations, which can effectively improve the running speed of the algorithm, avoiding more parameter settings and the excessive consumption of computing resources.

The rest of this paper is organized as follows. Section 2 introduces some basic concepts of dynamic multi-objective optimization problems and TradaBoost. Section 3 presents the proposed clustering difference strategy and its combination with transfer learning for solving dynamic multi-objective evolutionary optimization problems. Section 4 presents the experimental setup and results, and discusses the comparison with five other typical dynamic multi-objective optimization algorithms. Section 5 concludes this paper and presents an outlook for future research directions.

## 2. Background

### 2.1. Dynamic Multi-Objective Optimization Problems

The mathematical form of dynamic multi-objective optimization of DMOPs is as follows [37]:

$$\begin{cases} Minimize\ F(x,t) = \langle f_1(x,t), f_2(x,t), \ldots, f_m(x,t) \rangle \\ s.t. g_i(x,t) \leq 0, i = 1, 2, \ldots, p \\ h_j(x,t) = 0, j = 1, 2, \ldots, q \end{cases} \tag{1}$$

where $x = \langle x_1, x_2, \ldots, x_n \rangle$ is the decision vector, $t$ is a time or environment variable. $f_i(x,t) : \Omega \to \mathbb{R}(i = 1, \ldots, M)$ and $\Omega = [L_1, U_1] \times [L_2, U_2] \times \ldots \times [L_n, U_n]$ are the lower and upper bounds of the $i$-decision variable, respectively. $g_i(x,t) \leq 0, i = 1, 2, \ldots, p$ is the $i$-th inequality constraint, $h_j(x,t) = 0, j = 1, 2, \ldots, q$ is the $j$-th equality constraint. The purpose of solving the DMOP is to find a set of solutions in different times or environments, so that all objectives are as small as possible.

**Definition 1** (Dynamic Decision Vector Domination [38]). *At time t, the decision vector $x_1$ Pareto dominates another vector $x_2$, expressed as $x_1 \succ x_2$, if and only if*

$$\begin{cases} f_i(x_1,t) \leq f_i(x_2,t) & \forall i = 1, \ldots, m \\ f_i(x_1,t) \leq f_i(x_2,t) & \exists i = 1, \ldots, m. \end{cases} \tag{2}$$

**Definition 2** (Dynamic Pareto-Optimal Set (DPS) [39]). *If a decision vector $x^*$ at time t satisfies*

$$DPS = \{x^* \in \Omega | \exists x \in \Omega, x \succ_t x^*\} \tag{3}$$

*For a fixed time window t and a decision vector $x^* \in \Omega$, a decision vector $x^*$ is said to be non-dominant if no other decision vector $x \in \Omega$ dominates $x^*$, and the dynamic Pareto-optimal set (DPS) is the set of all non-dominated solutions in the decision space.*

**Definition 3** (Dynamic Pareto-Optimal Front (DPF) [39]). *DPF is the set of the corresponding objective vectors of the DPS, and*

$$DPF = \{y^* | y^* = F(x^*,t), x^* \in DPS\} \tag{4}$$

Algorithm 1 is the main framework of DMOEA. After initializing the current generation of the population, the algorithm employs several strategies to respond to the environment when it changes. The initialized population is updated with the effective strategy, and the time window t is incremented by 1 to represent the next environmental change. In the next step, the i-th multi-objective problem is optimized for generation using a multi-objective evolutionary algorithm. SMOEA uses the updated population as the initial population. Finally, repeat the process if the stopping condition is not met.

---

**Algorithm 1:** The main frame of DMOEA.

---

**Input:** The number of generations: $g$; the time window: $t$;
**Output:** Optimal solution $x^*$ at every time step;
Initialize population $POP_0$;
**While** stop criterion is not met do
　　**if** change is detected, then
　　Update the population using some strategies: reuse memory, tune parameters, or
predict solutions;
　　　$t = t + 1$;
　　**end if**
Optimize population with an MOEA for one generation and get optimal solution $x^*$;
**end while**
$g = g + 1$;
**return** $x^*$

---

### 2.2. TradaBoost

This paper adopts a method called TradaBoost to meet the requirement of DMOP. TradaBoost is evolved from the Adaboost algorithm, but the Adaboost algorithm, like most traditional machine learning algorithms [40], assumes that the data of the training set and the test set are the same distribution. For migration learning, this assumption is not true. In addition, for the part of the data in the training set that is different from the data in the test set, it will directly lead to a decline in the prediction effect. The TradaBoost algorithm adds weight to each training set sample, and uses the weight to weaken the test set data with different distributions, thereby improving the effect of the model. In each iterative training, if the model misclassifies a source domain sample, then this sample may have a large gap with the target domain sample, so the weight of this sample needs to be reduced. By multiplying the sample by a weight between 0 and 1, through the influence of the weight value, in the next iteration, the influence of this sample on the classification model will be reduced. After a series of iterations, the weights of samples in the source domain that is similar to the target domain or helpful to the classification of the target domain will increase, while the weights of other source domains will decrease.

If there are similarities between multiple source domain datasets and target datasets, in this case, you can try to use multiple source domain datasets to help the learning of the target dataset. More data can be obtained through the above method, so that the relationship between the source data and the target data becomes closer, the transfer process easier, and the classification effect more accurate.

Multi-source TradaBoost assumes that the source training data come from different source domains. In each iteration, the source domain most relevant to the target domain is selected to train a weak classifier, and finally, a strong classifier is obtained. This method can ensure that the transferred knowledge is most relevant to the target task, and through continuous learning, the TradaBoost algorithm can obtain a more accurate classifier for the target domain samples.

## 3. Proposed TCD-DMOEA

This section details the transfer learning based on clustering difference for the dynamic multi-objective optimization algorithm (TCD-DMOEA). Figure 2 describes the process of TCD-DMOEA. Specifically, first of all, the framework of the algorithm is outlined. Then, the specific process of the clustering type strategy is described. Finally, we analyze the calculation complexity of the strategy.

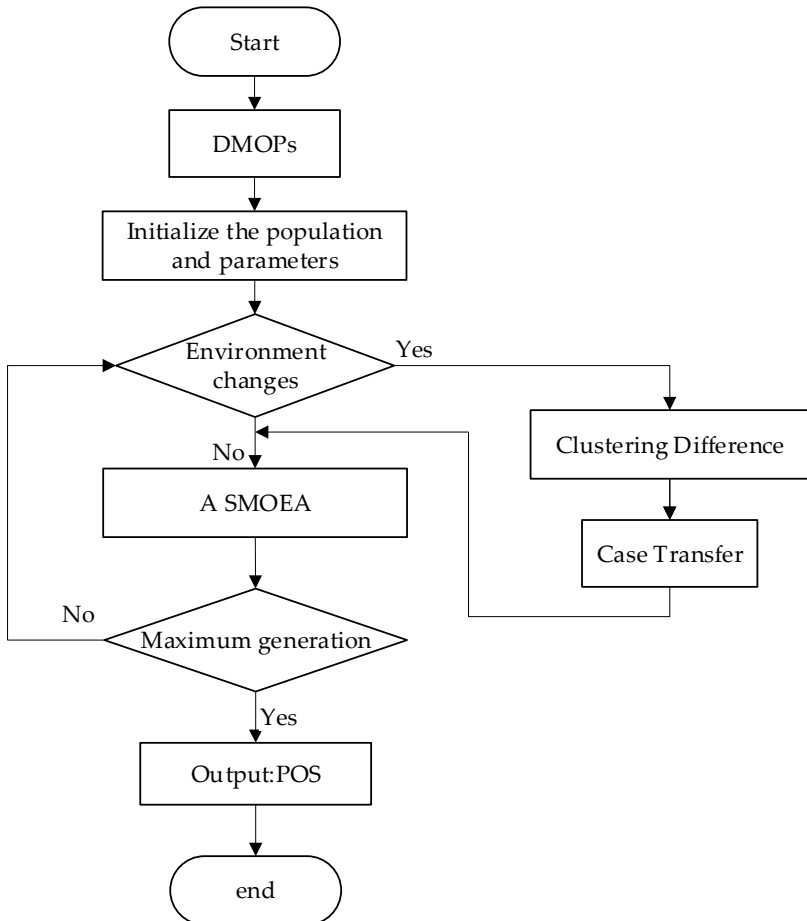

**Figure 2.** Procedure of TCD-DMOEA.

### 3.1. Overall Framework

The main framework of TCD-DMOEA is introduced in Algorithm 2. When an environmental change is detected, in the first two changes, evolution is performed using SMOEA. In subsequent changes, the clustering difference strategy is used to process the target domain (see Algorithm 3), and then the transfer learning prediction model (see Algorithm 4) is used to process high-quality individuals as the initial population for subsequent iterations.

---

**Algorithm 2:** TCD-DMOEA.

---

**Input:** The dynamic optimization problem $F_t(x)$, a static multi-objective optimization algorithm **SMOEA**;
**Output:** The POS of the $F_t(x)$ at the different moments;
Initialization;
$POS_t = \mathbf{SMOEA}(F_t(x))$;
Generate randomly dominated solutions $P_t{}'$;
    **while** *the environment has changed*, **do**
    $t = t + 1$;
    $D_S^{t-1} = POS_{t-1} \cup P'_{t-1}$;
    $D_T^t = \mathbf{Processing}(F_t(x))$;
    $iPop = \mathbf{Case\text{-}Transfer}(D_S^{t-1} \cup D_T^t)$;
    $POS_t = \mathbf{SMOEA}(iPop, F_t(x))$;
    Generate randomly dominated solutions $P_t{}'$;
    **return** $POS_t$
**end while**

---

---

**Algorithm 3:** Processing.

---

**Input:** The current population $P_T$; the number of individuals in population, $N$;
**Output:** The predicted population $PP$
Initialize the random population $P_T$ and evaluate the initial population $P_T$;
Change detection $(P_T)$;
**if** change is detected, **then**
    **while** the maximum number of iterations is not reached, **do**
        **for** $i = 1 : N$ **do**
        Use K-means algorithm to cluster the population $P$ into 5 clusters;
        Calculate the centroid $C_i{}^T$ of each cluster;
        Calculate $C_i{}^{T+1}$ using Formula (8);
        **end for**
    $P_{T+1} = \{C_i{}^{T+1}\}$;
    **end while**
**end if**
$PP = P_{T+1}$;
**return** $PP$

---

---

**Algorithm 4:** Case transfer.

---

**Input:** The two labeled sets $D_S$ and $D_T$, and unlabeled data set $D$, a based learning algorithm **Learner**, and the maximum number of iterations $N$;
**Output:** The initial population *initPop*;

Initialize the initial weight vector $\omega_1(x) = \begin{cases} \frac{1}{|D_S|}, x \in X_S \\ \frac{1}{D_T}, x \in X_T \end{cases}$;
    **for** $i = 1 \ to \ N$ **do**
    set $P^t$ according to (13);
    Call Learner, providing it the combined training set $D$ with the distribution over $D$. Then, get back a hypothesis $h_t : X \to Y$;
    Calculate $\varepsilon_t$ according to (9);
    Set $\beta_t$, Update the weight vector $\omega_i^{t+1}$ according to (10);
    **end for**
Get $h_{f(x)}$ according to (11);
Sample solutions $x_{test}$ at the current environment;
**return** $initPop = \left\{ x \middle| h_{f(x)} = +1, \ x \in x_{test} \right\}$

---

### 3.2. Processing of Target Domain

The output of the target domain processing stage is the predicted population. The purpose of generating predicted population is to reduce the possibility of negative transfer in subsequent transfer learning. According to the characteristics of transfer learning, negative transfer can be improved by increasing the amount of effective source domain knowledge or reducing the data distribution difference between neighbors. Through the clustering difference strategy, the characteristics of the source domain and the target domain data are first extracted to reduce the difference between the data, and then the knowledge transfer is performed. At the same time, the population quality is improved through the clustering difference strategy to increase the amount of effective source domain knowledge.

In the dynamic environment, the objective function changes with time, but there is a certain relationship between the two objectives before and after the change. Therefore, the optimal solution information before the change can be used to predict the distribution of the next solution. First, the population is divided into five categories using the k-means algorithm, and the centroids of these five categories are calculated separately. Then, the first-order difference is used to predict the next corresponding centroid, and these centroids are formed into a predicted population. Figure 3 shows the process of prediction of clustering differential strategies.

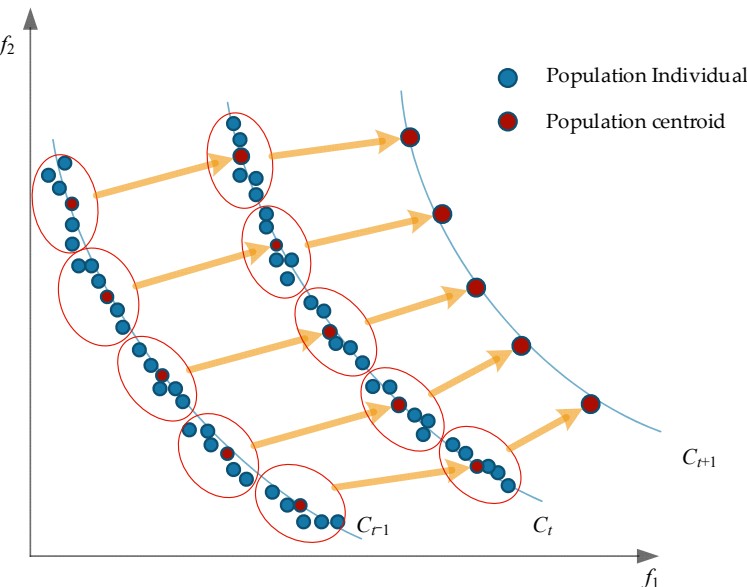

**Figure 3.** Classification of clustering dynamics.

Algorithm 3 gives the pseudo code of target domain processing. The basic principle of K-means algorithm is: assuming a given data sample $X$, contains $n$ objects $X = \{X_1, X_2, X_3, \ldots, X_n\}$, each of these objects has $m$-dimensions attributes. The goal of the K-means algorithm is to cluster $n$ objects into a specified $k$-class cluster based on similarity between objects. Each object belongs to only one of the class clusters with the smallest distance to the center of the class cluster. For K-means, $k$ cluster centers $\{C_1, C_2, C_3, \ldots, C_k\}, 1 < k \leq n$ need to be initialized first; the Euclidean distance from each object to the center of each cluster is calculated as shown in the following Formula (5):

$$dis(X_i, C_j) = \sqrt{\sum_{t=1}^{m} (X_{it} - C_{jt})^2} \tag{5}$$

In the above equation, $X_i$ represents the $i$-th object $1 \leq i \leq n$, $C_j$ represents the center of the $j$-th cluster $1 \leq j \leq k$, $X_{it}$ represents the $t$-property of the $i$-th object, $1 \leq t \leq m$, $C_{jt}$ represents the $t$-th attribute of the $j$-th cluster center.

The distance from each object to each cluster center is compared sequentially, and the objects are assigned to the cluster of the nearest cluster center, resulting in $k$ class clusters $\{S_1, S_2, S_3, \ldots, S_k\}$.

The K-means algorithm defines the prototype of the class cluster with the center, which is the average of all objects in the class cluster in each dimension, and its calculation process is shown in Formula (6):

$$C_t = \frac{\sum_{X_i \in S_l} X_i}{|S_l|} \tag{6}$$

where $C_l$ represents the center of the $l$-th cluster, $1 \leq l \leq k$, $S_l$ represents the number of objects in the $l$-th class cluster, $X_i$ represents the $i$-th object in the $l$-th class cluster, $1 \leq i \leq |S_l|$. The population is divided into five categories according to the K-means principle above, and the centroid $C_i^T (i = 1, 2, \ldots, 5)$ of each cluster is calculated after clustering.

The first-order differences are then used to derive the centroids $C_i^T (i = 1, 2, \ldots, 5)$ of each cluster at the next moment. $P_T$ is the DPS obtained by the time window $T$, then $C_i^T$ can be calculated by the following Formula (7):

$$C_i^T = \frac{1}{|P_T|} \sum_{x \in P_T} x \tag{7}$$

where $C_i^{T+1}$ represents the centroid of each cluster in the next time window *T+1*, as obtained by Formula (8):

$$C_i^{T+1} = C_i^T + C_i^T - \overrightarrow{C_i^{T-1}} \tag{8}$$

where $C_i^{T+1}$ will constitute a predicted population.

### 3.3. Transfer Learning

After the predicted population is generated, the source and target domains for transfer learning are specified. Figure 4 describes the instance migration program of TCD-DMOEA. Step 1: Treat the target domain; step 2: The solution in which step 1 is processed and the previous solution (not just the nondominated solutions) are entered into the TradaBoost algorithm. Step 3: Use the TradaBoost algorithm to generate a strong classifier $h_f$. Step 4: Enter $h_f$ multiple individuals generated by the current time. Step 5: Individuals recognized as "good" by the classifier $h_f$ will form a high-quality initial group.

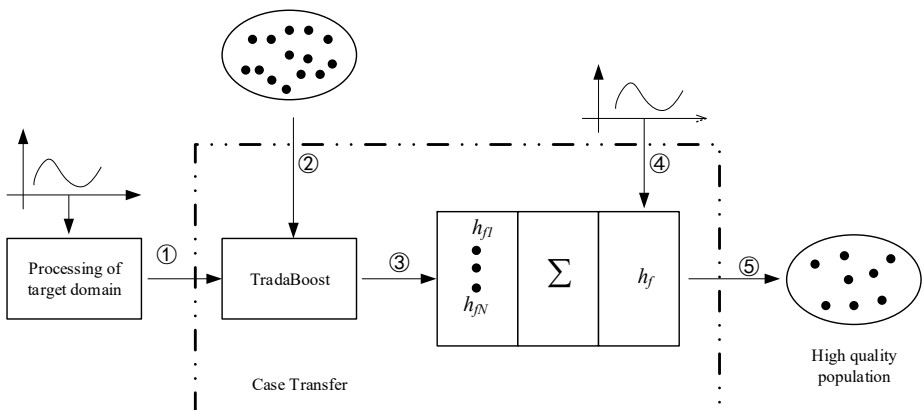

**Figure 4.** Schematic of procedure case transfer.

Here, the TradaBoost algorithm is mainly used to realize the transfer, increase the weight of each training set sample using the weight to weaken the test set data of those different distributions, and then improve the effect of the model. In each iteration of training, if the model misclassifies a sample of the source domain, the sample may have a large gap with the sample of the target domain, so the weight of the sample needs to be reduced. By multiplying the sample by a weight between 0 and 1, by the effect of the weight value, in the next iteration, the effect of the sample on the classification model will be reduced, and after a series of iterations, the weight of the samples in the source domains that are similar to the target domain or helpful to the classification of the target domain will be increased, while the weights of other source domains will be reduced. When the training is complete, the classifier can recognize randomly generated solutions in the current environment and select those individuals identified as "good" by the classifier as the initial population. The specific algorithm pseudo code is described in detail in Algorithm 4.

Algorithm 4 show the main program of the proposed TCD-DMOEA method. During the transfer process, the target domain $D_T$ is the predicted population, and the source domain $D_S$ is the solution obtained in the past environment. These solutions are then labeled with $c(x) : x \in D_T \cup D_S \to y, y \in \{+1, -1\}$. For each domain, the non-dominant solution is labeled +1, and the dominant solution is labeled −1.

Call Learner, according to the combined training data D and the weight distribution $P^t$ on D and the unlabeled data *s*, to obtain a classifier $h_t : X \to Y$ in $S$; statistics on the error rate of $h_t$ on $D_t$ by Formula (9):

$$\varepsilon_t = \sum_{i=n+1}^{n+m} \frac{\omega_i^t |h_t(x_i) - c(x_i)|}{\sum_{i=n+1}^{n+m} \omega_i^t} \tag{9}$$

If $\varepsilon_t > 0.5$ then the algorithm terminates; set up $\beta_t = \varepsilon_t / (1 - \varepsilon_t)$, $\beta = 1 / (1 + \sqrt{2 \ln n / N})$, sets the new weight vector by Formula (10):

$$\omega_i^{t+1} = \begin{cases} \omega_i^t \beta^{|h_t(x_i) - c(x_i)|}, i = 1, \ldots, m \\ \omega_i^t \beta^{-|h_t(x_i) - c(x_i)|}, i = m+1, \ldots, m+n \end{cases} \tag{10}$$

Finally, output the final classifier by Formula (11):

$$h_{f(x)} = \begin{cases} 1, \sum_{t=[N/2]}^{N} \ln\left(\frac{1}{\beta_t}\right) h_t(x) \geq 1/2 \sum_{t=[N/2]}^{N} \sum_{t=[N/2]}^{N} \ln\left(\frac{1}{\beta_t}\right) \\ 0, \text{other} \end{cases} \tag{11}$$

During the pre-build phase of the source domain, the output is the predicted population. In this approach, the main purpose of transfer learning after preprocessing is to reduce the possibility of negative transfer.

Given the sample with an initial weight vector $\omega^1 = \left(\omega_1^1, \ldots, \omega_{n+m}^1\right)$, where $\omega_i^1$ is obtained from Formula (12):

$$\omega_i^1 = \begin{cases} \frac{1}{m}, i = 1, \ldots, m \\ \frac{1}{n}, i = m+1, \ldots, m+n \end{cases} \tag{12}$$

At the same time, set $P^t$ to satisfy Formula (13).

$$P^t = \frac{\omega^t}{\sum_{i=1}^{n+m} \omega_i^t} \tag{13}$$

### 3.4. Computational Complexity Analysis

This section analyzes the computational complexity of TCD-DMOEA at one iteration. According to Algorithm 2, the main calculation of TCD-DMOEA comes from the following aspects. (1) The complexity of the target domain preprocessing process mainly lies in the use of K-means clustering and first-order difference to predict the non-dominated solution at the next moment. K-means clustering requires $O(Inkm)$ calculation, where $m$ is the number of element fields, $n$ is the amount of data, $I$ represents the number of iterations, and $k$ is the number of clusters. Generally, $I$, $k$, and $m$ can be considered as constants, so the computational complexity can be simplified to: $O(n)$. The first difference requires $O(n)$ computation. The computational complexity of the target domain preprocessing stage is $O(n)$. (2) The transfer learning stage uses SVM as the basic classifier, and the SVM classifier costs $O(NS^2 d)$ to obtain a strong classifier, where $S$ is the overall size, $N$ is the number of iterations, and $d$ is the dimension of the decision variable. To sum up, the computational complexity of TCD-DMOEA in this work is $O(NS^2 d)$.

## 4. Experiments

### 4.1. Test Problems and Performance Indicators

Our experiment was divided into two parts: the first part demonstrated the convergence and distribution uniformity of TCD-DMOEA by comparing it with several popular dynamic multi-objective algorithms. In the second part, through the comparison with Tr-DMOEA [28], it was possible to observe a reduction in the running time, effectively reducing the possibility of negative transfer. The entire experiment was based on MATLAB R2019b, running in a Windows 10 Pro.

The strategy proposed by the experimental setup was mainly used during the initialization stage of the algorithm. A suite of preprocessing and transfer learning techniques allowed us to obtain a high-quality population adapted to the current environment, and evolution to obtain the optimal solution after modification. Theoretically, the target domain generated by the clustering difference strategy is more similar to the source domain, and the initial population improved by the transfer learning method is more adaptable to the

changing environment, so as to obtain a solution closer to the actual DPF. To verify the effectiveness of the method, this section used 20 test functions and two related metrics to measure the convergence and uniformity of the algorithm, while using the running time to evaluate the negative transfer possibility of the algorithm.

The 20 test functions used in this section were from CEC 2018: (1) DF function [41], and (2) F function [42]. The DF Benchmark Suite contains 14 questions (DF1-DF14) and the F Benchmark Suite contains six questions (F5–F10). The DF function is a diverse and unbiased benchmark problem, covering various attributes that represent various real scenes, such as time-dependent PF/PS geometry, irregular PF shape, disconnection, knee, etc. F5–F8 in F function have nonlinear correlations among decision variables. The PSs of F5–F7 are 1-D curves, and the PSs of F8 are 2-D surfaces. In F9, the environment changes smoothly in most cases. Sometimes, Pareto sets jump from one region to another. In F10, the geometry of two consecutive PFs is completely different.

In 20 test benchmarks, the time parameter $t$ was used here and defined as $t = (1/n_t) \lfloor (\tau/\tau_t) \rfloor$, where $\tau$, $n_t$, and $\tau_t$ represented the maximum number of iterations, the severity of the change, and the frequency of the change, respectively, as described in Table 1. Different kinetic parameters were set for the experiment. We set different variation severities, frequencies of change, and numbers of iterations so that each function could iterate 20 times, and the entire population size was set to 200, meaning that 200 solutions could be generated during evolution. In addition, the k value for K-means was set to 5. This experiment chose RM-MEDA [43] as the SMOEA optimizer of TCD-DMOEA. In this study, the following metrics were used to evaluate the performance of different algorithms.

**Table 1.** The dynamic parameters.

| Settings | Change Severity $n_t$ | Change Frequency $\tau_t$ | Maximum Iteration $\tau$ |
|---|---|---|---|
| S1 | 10 | 5 | 100 |
| S2 | 5 | 10 | 200 |
| S3 | 10 | 10 | 200 |

(1) Inverted generational distance (IGD [44–46]): IGD evaluates the convergence and diversity of algorithms by measuring the proximity between the real Pareto frontier and the Pareto frontier obtained by the algorithm, and the definition of the IGD indicator is calculated by Formula (14), where $d$ is calculated by Formula (15).

$$IGD(PF^*{}_t, PF_t) = \frac{\sum_{v \in PF_{F_t}^*} d(v, PF_t)}{|PF_t^*|} \tag{14}$$

$$d(v, P_t) = \min_{u \in P_t} \sqrt{\sum_{j=1}^{m} \left( f_j^v - f_j^u \right)^2} \tag{15}$$

where $PF^*{}_t$ is the standard POF of the $t$-moment, it is the POF obtained by the $t$-moment algorithm, and $d$ is the Euclidean distance between the individual $v$ on the $PF^*{}_t$ and the individual closest to $v$ in $PF_t$. It can be seen that the evaluation method of IGD is for each individual in the standard POF. $PF_t$ is used to find the closest point to it in the POF. $PF_t$ is obtained by the algorithm, the Euclidean distance between them is calculated, and then all the Euclidean distances are summed and the average taken, so IGD can not only evaluate the proximity between $PF^*{}_t$ and $PF_t$, but also evaluate the distribution of individuals in $PF_t$; the smaller the IGD value, the better the convergence of the POF obtained by the algorithm, and the more uniform the distribution.

The MIGD [42,47] indicator is a variant of IGD and is defined as the average of IGD values over certain time steps in operation. The MIGD value is calculated by Formula (16):

$$MIGD = \frac{1}{|T|}\sum_{t \in T} IGD(t) \tag{16}$$

where *IGD(t)* represents the IGD value at time *t*, *T* is a set of discrete time points in operation, and $|T|$ is the cardinality of *T*.

(2) Maximum coverage (MS [31,48]): MS measures the extent to which the Pareto front obtained by the algorithm covers the standard Pareto front. The larger the MS, the better the performance of the algorithm. The MS value is calculated by Formula (17):

$$MS_t = \sqrt{\frac{1}{m}\sum_{i=1}^{m}\left[\frac{\min\left[\overline{PF_{ti}}, \overline{PF_{ti}^*}\right] - \max\left[\underline{PF_{ti}}, \underline{PF_{ti}^*}\right]}{\overline{PF_{ti}^*} - \underline{PF_{ti}^*}}\right]^2} \tag{17}$$

where $PF_t^*$ is the standard Pareto frontier of *t*-moment, and $PF_t$ is the POF obtained by the *t*-moment algorithm. Where $\overline{PF_{ti}}$ and $\underline{PF_{ti}}$ are the maximum and minimum values of the *i*-th target of the POF obtained by the *t*-moment algorithm, respectively, $\overline{PF_{ti}^*}$ and $\underline{PF_{ti}^*}$ are the maximum and minimum values of the *i*-th target of the real Pareto frontier at the *t*-moment.

### 4.2. Performance Comparison with Other Algorithms

In this section, performance comparison experiments are performed. The above indicators MIGD and MS are used, and some algorithms are compared with the algorithm proposed in this paper. Firstly, the MIGD values of six algorithms are compared, proving the proposed algorithm's effectiveness. These six algorithms include a dynamic multi-objective optimization algorithm (TCD-DMOEA) that combines clustering difference and transfer learning, a dynamic multi-objective optimization algorithm (Tr-DMOEA) that combines only transfer learning methods [28], a dynamic NSGA-II algorithm (DNSGA-II-A, DNSGA-II-B) [49], PPS [42], and MOEA based on Kalman's predictions (KF-DMOEA) [50]. A comparison of TCD-DMOEA and Tr-DMOEA was conducted to prove the performance of our proposed strategy. According to the difference in performance index and running time, it could be determined that the clustering difference strategy proposed in this paper can not only maintain good convergence and diversity, but also effectively reduce the possibility of negative transfer.

Figure 5 plots the IGD values obtained by different algorithms after each change. These curves show that under 20 different test functions, the curve obtained by the method was basically at the bottom, and the curve of the method fluctuated less, which meant that the method is not only better performing, but also more stable.

The statistical results of MIGD and MS values that were run 20 times are shown in Tables 2 and 3, respectively. Table 2 shows the MIGD values for six algorithms in three different configurations. Bold words in the table indicate that the algorithm had the best diversity on this benchmark, and the last column represents the "winner" in this comparison. It can be seen from Table 2 that TCD-DMOEA obtained 44 of the 60 best results, accounting for 73.3%. KF-DMOEA achieved five best results, PPS achieved three best results, and DNSGA-II-A and DNSGA-II-B achieved six and two best results, respectively. Specifically, TCD-DMOEA performed well at most test functions in all dynamic test setups, and DNSGA-II-A achieved better convergence than TCD-DMOEA on DF9 and DF12. The value of PPS was obviously superior to that of other algorithms on DF11.

Table 3 shows the MS values for six comparison algorithms, and it is clear that TCD-DMOEA obtained 38 out of 60 best results, KF-DMOEA obtained only one best result, PPS obtained five best results, and Tr-DMOEA obtained nine best results. DNSGA-II-A and DNSGA-II-B achieved four and three best results, respectively. Specifically, TCD-

DMOEA performed well at most test functions in all dynamic test setups, with Tr-DMOEA performing better on F8 and DNSGA-II-B performing better on DF10.

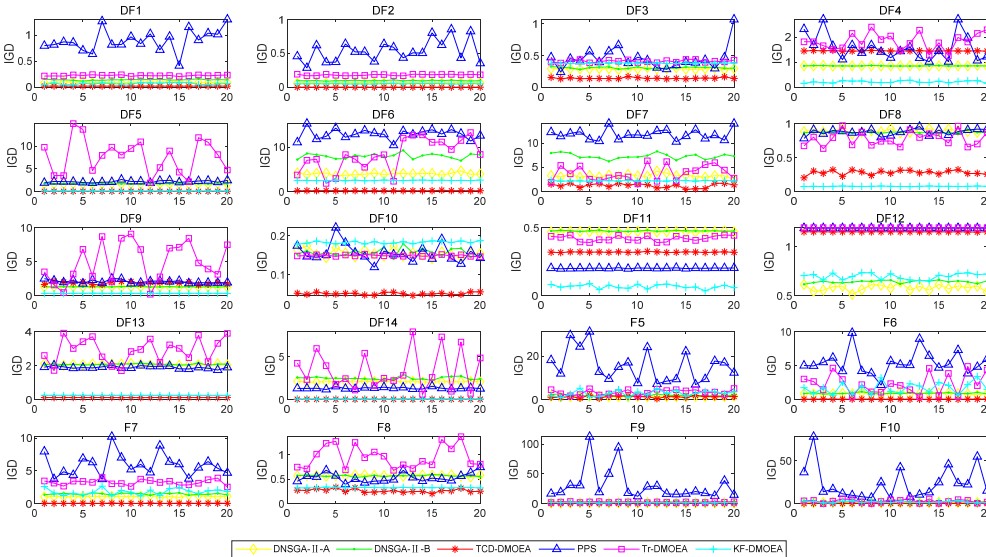

**Figure 5.** IGD values of six algorithms under S2 configuration.

The above experimental results show that TCD-DMOEA could obtain a set of solutions with good convergence and diversity for most test problems. However, it did not perform well enough on some reference functions, such as DF9, DF11, and DF12. TCD-DMOEA also had some shortcomings, which may have been due to the large variation in the POS of these problems, so it was difficult to accurately predict in the target domain processing stage, and the final output of the population quality was poor.

At the same time, it can be found from Table 2 that except for F8, F9 and F10, TCD-DMOEA achieved good performance on the F test function set in most cases. These three test functions were characterized by POS jumps from one region to another, and the geometry of the two consecutive POFs was completely different. This resulted in a low degree of similarity between the source domain and the target domain, which ultimately increased the likelihood of negative transfer.

*4.3. Running Speed*

The most obvious feature that reduces the possibility of negative transfer is reduced running time, so the running times of different algorithms are compared and shown in Table 4. Table 4 shows that the running time of TCD-DMOEA was the shortest among the six algorithms. This shows that the proposed clustering difference strategy was very effective. TCD-DMOEA ran much faster than Tr-DMOEA. The important difference is that Tr-DMOEA mapped the different distributions that the solution obeys at varying moments to a new potential space through the domain adaptation method, and then found the solution in the potential space, and the construction of the potential space required a huge amount of resources. However, TCD-DMOEA improved the similarity between the source domain and the target domain by preprocessing the population and distinguished the quality of the population through classifiers, which significantly shortened the running time. Figure 6 is a histogram of the running time of several other algorithms except Tr-DMOEA, through which the running time of TCD-DMOEA can be visually compared, and it was found that the running time of TCD-DMOEA was the shortest, while the running times of other algorithms were not much different. The above experimental results show that the proposed TCD-DMOEA method not only improved the running speed of the algorithm, but also greatly improved the quality of the Pareto optimal solution set, resulting in a better state of convergence and distribution for the algorithm.

**Table 2.** The MIGD values of six comparative algorithms with different dynamic test settings.

| Functions | Settings | DNSGA-II-A | DNSGA-II-B | TCD-DMOEA | PPS | Tr-DMOEA | KF-DMOEA | Winner |
|---|---|---|---|---|---|---|---|---|
| DF1 | S1 | 0.12549 | 0.18466 | **0.01825** | 0.79948 | 0.16762 | 0.18591 | TCD-DMOEA |
|  | S2 | 0.14103 | 0.14997 | **0.02630** | 1.00778 | 0.22634 | 0.20192 | TCD-DMOEA |
|  | S3 | 0.06910 | 0.07078 | **0.01770** | 0.34875 | 0.17491 | 0.15986 | TCD-DMOEA |
| DF2 | S1 | 0.07984 | 0.11042 | **0.00577** | 0.56862 | 0.1088 | 0.12259 | TCD-DMOEA |
|  | S2 | 0.09846 | 0.09856 | **0.00521** | 0.55064 | 0.1808 | 0.14677 | TCD-DMOEA |
|  | S3 | 0.04331 | 0.05206 | **0.00519** | 0.33104 | 0.1315 | 0.10525 | TCD-DMOEA |
| DF3 | S1 | 0.36036 | 0.31444 | **0.06176** | 0.46092 | 0.315 | 0.38322 | TCD-DMOEA |
|  | S2 | 0.31344 | 0.30096 | **0.22758** | 0.56408 | 0.4117 | 0.36154 | TCD-DMOEA |
|  | S3 | 0.34277 | 0.35370 | **0.05804** | 0.28151 | 0.3406 | 0.37636 | TCD-DMOEA |
| DF4 | S1 | 1.33552 | 1.29683 | **0.92736** | 4.09048 | 1.5689 | 1.20196 | TCD-DMOEA |
|  | S2 | 0.85006 | **0.85114** | 1.45397 | 1.78053 | 1.8344 | 1.98927 | DNSGA-II-B |
|  | S3 | 1.29904 | 1.28637 | **1.03469** | 2.13623 | 1.5869 | 1.98173 | TCD-DMOEA |
| DF5 | S1 | 0.09852 | 0.18511 | **0.02306** | 0.36546 | 2.3146 | 1.35092 | TCD-DMOEA |
|  | S2 | 1.67522 | 1.68125 | **0.02317** | 2.00282 | 2.6205 | 3.26962 | TCD-DMOEA |
|  | S3 | 0.05721 | 0.07235 | **0.02274** | 0.25696 | 2.4713 | 3.30728 | TCD-DMOEA |
| DF6 | S1 | 5.50747 | 7.96408 | **0.46630** | 11.8778 | 5.5982 | 6.26992 | TCD-DMOEA |
|  | S2 | 7.32493 | 8.29557 | **0.16436** | 12.4986 | 7.2692 | 8.57652 | TCD-DMOEA |
|  | S3 | 2.81471 | 3.45856 | **0.21433** | 5.66312 | 6.7379 | 6.55881 | TCD-DMOEA |
| DF7 | S1 | 4.94594 | 8.21914 | **0.51930** | 10.0823 | 8.8403 | 7.45962 | TCD-DMOEA |
|  | S2 | 7.43732 | 7.84210 | **0.35659** | 11.3676 | 4.2387 | 8.67889 | TCD-DMOEA |
|  | S3 | 2.19079 | 3.20643 | **1.13082** | 5.96671 | 4.0323 | 8.9454 | TCD-DMOEA |
| DF8 | S1 | 0.83967 | 0.88308 | **0.06331** | 0.87168 | 0.78817 | 1.10952 | TCD-DMOEA |
|  | S2 | 0.86939 | 0.85910 | **0.29914** | 0.85785 | 0.7993 | 1.41749 | TCD-DMOEA |
|  | S3 | 0.88772 | 0.89634 | **0.05962** | 0.86689 | 0.8026 | 1.64865 | TCD-DMOEA |
| DF9 | S1 | **1.44013** | 1.48968 | 2.03433 | 1.94736 | 2.5958 | 2.81085 | DNSGA-II-A |
|  | S2 | **1.26994** | 1.27597 | 1.67976 | 2.24801 | 2.7079 | 3.17112 | DNSGA-II-A |
|  | S3 | **1.59835** | 1.60138 | 2.33938 | 1.66393 | 2.3714 | 3.28462 | DNSGA-II-A |

**Table 2.** *Cont.*

| Functions | Settings | DNSGA-II-A | DNSGA-II-B | TCD-DMOEA | PPS | Tr-DMOEA | KF-DMOEA | Winner |
|---|---|---|---|---|---|---|---|---|
| DF10 | S1 | 0.14762 | 0.14492 | **0.05083** | 0.16937 | 0.14870 | 0.23669 | TCD-DMOEA |
| | S2 | 0.15983 | 0.14948 | **0.05854** | 0.15284 | 0.1493 | 0.24198 | TCD-DMOEA |
| | S3 | 0.13147 | 0.12085 | **0.05061** | 0.12691 | 0.1194 | 0.21441 | TCD-DMOEA |
| DF11 | S1 | 0.40398 | 0.40980 | 0.19190 | **0.12101** | 0.38975 | 0.26247 | PPS |
| | S2 | 0.47354 | 0.47851 | 0.32070 | **0.20203** | 0.4178 | 0.19754 | PPS |
| | S3 | 0.39052 | 0.39487 | 0.20528 | **0.11272** | 0.3331 | 0.1851 | PPS |
| DF12 | S1 | **0.59895** | 0.64389 | 1.15032 | 1.18370 | 1.19331 | 0.91213 | DNSGA-II-A |
| | S2 | **0.64129** | 0.65550 | 1.14949 | 1.18769 | 1.1923 | 1.25910 | DNSGA-II-A |
| | S3 | **0.61309** | 0.68070 | 1.14954 | 1.18368 | 1.19 | 0.989 | DNSGA-II-A |
| DF13 | S1 | 0.58572 | 0.66171 | **0.17489** | 0.24456 | 3.62620 | 3.37829 | TCD-DMOEA |
| | S2 | 2.07238 | 2.07951 | **0.11530** | 1.78561 | 2.8032 | 3.55246 | TCD-DMOEA |
| | S3 | 0.53317 | 0.55064 | **0.18022** | 0.25370 | 2.7312 | 1.4413 | TCD-DMOEA |
| DF14 | S1 | 0.17673 | 0.61880 | **0.02471** | 0.18437 | 1.6727 | 2.29643 | TCD-DMOEA |
| | S2 | 2.42269 | 2.37278 | **0.03635** | 1.35517 | 1.8333 | 2.34678 | TCD-DMOEA |
| | S3 | 0.15027 | 0.59957 | **0.02626** | 0.08840 | 1.8257 | 1.90653 | TCD-DMOEA |
| F5 | S1 | 1.79444 | 2.36932 | **0.19901** | 5.95315 | 2.8026 | 4.90052 | TCD-DMOEA |
| | S2 | 1.78814 | 1.58863 | **1.04138** | 14.2358 | 3.6919 | 5.24772 | TCD-DMOEA |
| | S3 | 0.85822 | 1.01156 | **0.11964** | 2.79008 | 2.6592 | 6.85162 | TCD-DMOEA |
| F6 | S1 | 1.16622 | 1.24886 | **0.33671** | 2.69581 | 1.2349 | 4.21603 | TCD-DMOEA |
| | S2 | 0.82511 | 0.84377 | **0.03847** | 4.49749 | 2.4094 | 3.94881 | TCD-DMOEA |
| | S3 | 0.86284 | 0.83938 | **0.13659** | 2.26413 | 1.3095 | 1.32448 | TCD-DMOEA |
| F7 | S1 | 1.69802 | 1.87903 | **0.07907** | 4.18370 | 1.4295 | 3.26674 | TCD-DMOEA |
| | S2 | 1.53415 | 1.57008 | **0.06860** | 10.9552 | 3.1593 | 1.94158 | TCD-DMOEA |
| | S3 | 0.90880 | 0.90245 | **0.05635** | 1.95524 | 1.327 | 1.6357 | TCD-DMOEA |
| F8 | S1 | 0.61626 | 0.58686 | 0.24926 | 0.89452 | 0.74194 | **0.23932** | KF-DMOEA |
| | S2 | 0.57003 | 0.57302 | **0.31896** | 0.61123 | 1.0615 | 0.32661 | TCD-DMOEA |
| | S3 | 0.49723 | 0.51284 | 0.29669 | 0.30842 | 0.7875 | **0.21715** | KF-DMOEA |

**Table 2.** *Cont.*

| Functions | Settings | DNSGA-II-A | DNSGA-II-B | TCD-DMOEA | PPS | Tr-DMOEA | KF-DMOEA | Winner |
|---|---|---|---|---|---|---|---|---|
| | S1 | 2.16322 | 3.43708 | 0.89799 | 16.9765 | 1.85947 | **0.72251** | KF-DMOEA |
| F9 | S2 | 2.79212 | 2.79170 | **0.24658** | 26.6893 | 2.6079 | 1.70666 | TCD-DMOEA |
| | S3 | 0.90504 | 1.72920 | 1.36171 | 8.46293 | 1.4721 | **0.82095** | KF-DMOEA |
| | S1 | 2.80464 | 3.23187 | 4.14096 | 10.3253 | 2.04876 | **0.69335** | KF-DMOEA |
| F10 | S2 | 1.89147 | 2.04139 | **0.17199** | 10.3688 | 2.7845 | 3.83634 | TCD-DMOEA |
| | S3 | 2.58958 | **2.52480** | 4.10541 | 6.39496 | 2.7327 | 8.66578 | DNSGA-II-B |

**Table 3.** The MS values of six comparative algorithms with different dynamic test settings.

| Functions | Settings | DNSGA-II-A | DNSGA-II-B | TCD-DMOEA | PPS | Tr-DMOEA | KF-DMOEA | Winner |
|---|---|---|---|---|---|---|---|---|
| | S1 | 0.87298 | 0.83399 | **0.99596** | 0.65263 | 0.9203 | 0.7995 | TCD-DMOEA |
| DF1 | S2 | 0.87466 | 0.87180 | **0.99608** | 0.64542 | 0.84163 | 0.7807 | TCD-DMOEA |
| | S3 | 0.92019 | 0.92503 | **0.99328** | 0.85485 | 0.88981 | 0.8143 | TCD-DMOEA |
| | S1 | 0.91976 | 0.90297 | **0.99738** | 0.52831 | 0.9473 | 0.8202 | TCD-DMOEA |
| DF2 | S2 | 0.91425 | 0.91246 | **0.99786** | 0.71981 | 0.91661 | 0.8029 | TCD-DMOEA |
| | S3 | 0.94907 | 0.94420 | **0.98863** | 0.74677 | 0.93941 | 0.833 | TCD-DMOEA |
| | S1 | 0.34843 | 0.38187 | **0.75188** | 0.44420 | 0.4701 | 0.23 | TCD-DMOEA |
| DF3 | S2 | 0.54510 | 0.54590 | 0.54075 | **0.61212** | 0.44961 | 0.2828 | PPS |
| | S3 | 0.31963 | 0.30727 | **0.94657** | 0.42035 | 0.61989 | 0.2195 | TCD-DMOEA |
| | S1 | 0.23576 | 0.24019 | 0.37760 | 0.29071 | **0.4383** | 0.2705 | Tr-DMOEA |
| DF4 | S2 | 0.33726 | 0.33918 | 0.28217 | **0.37772** | 0.24422 | 0.2985 | PPS |
| | S3 | 0.23191 | 0.23489 | **0.56408** | 0.28755 | 0.31761 | 0.308 | TCD-DMOEA |
| | S1 | 0.99550 | 0.99626 | 0.99991 | 0.99233 | **1** | 0.9426 | Tr-DMOEA |
| DF5 | S2 | 0.99769 | 0.99685 | **0.99988** | 0.99959 | 0.99865 | 0.956 | TCD-DMOEA |
| | S3 | 0.99790 | 0.99836 | 0.99993 | **0.99998** | 0.99900 | 0.9303 | PPS |
| | S1 | 0.89098 | 0.94927 | 0.99909 | 0.931395 | **1** | 0.7099 | Tr-DMOEA |
| DF6 | S2 | 0.99325 | 0.99724 | **0.99999** | 0.898478 | 0.632845 | 0.8682 | TCD-DMOEA |
| | S3 | 0.96554 | 0.98084 | **0.99962** | 0.966298 | 0.61565 | 0.75 | TCD-DMOEA |

**Table 3.** *Cont.*

| Functions | Settings | DNSGA-II-A | DNSGA-II-B | TCD-DMOEA | PPS | Tr-DMOEA | KF-DMOEA | Winner |
|---|---|---|---|---|---|---|---|---|
| DF7 | S1 | 0.9 | 0.93785 | 0.95660 | 0.920775 | **1** | 0.7155 | Tr-DMOEA |
| | S2 | **1** | **1** | 0.82791 | 0.9 | 0.691549 | 0.8441 | DNSGA-II-A |
| | S3 | 0.92498 | 0.94743 | **1** | 0.86527 | 0.66624 | 0.7515 | TCD-DMOEA |
| DF8 | S1 | 0.36573 | 0.35096 | **0.86352** | 0.37147 | 0.4501 | 0.3004 | TCD-DMOEA |
| | S2 | 0.35244 | 0.34334 | 0.63800 | 0.37548 | **0.71035** | 0.6078 | Tr-DMOEA |
| | S3 | 0.32656 | 0.33039 | **0.92433** | 0.32542 | 0.65714 | 0.4329 | TCD-DMOEA |
| DF9 | S1 | **0.85204** | 0.84753 | 0.35128 | 0.763924 | 0.8068 | 0.6775 | DNSGA-II-A |
| | S2 | 0.91461 | 0.92985 | 0.69664 | **0.92934** | 0.74104 | 0.7588 | PPS |
| | S3 | 0.76468 | 0.78770 | **0.83142** | 0.80924 | 0.64614 | 0.7012 | TCD-DMOEA |
| DF10 | S1 | 0.98810 | **1** | 0.99999 | 0.99426 | 0.9998 | 0.9502 | DNSGA-II-B |
| | S2 | 0.99999 | **1** | 0.99999 | 0.99559 | 0.99938 | 0.9927 | DNSGA-II-B |
| | S3 | 0.99151 | **1** | 0.99999 | 0.99842 | 0.99962 | 0.9125 | DNSGA-II-B |
| DF11 | S1 | 0.71851 | 0.71067 | **0.94698** | 0.94667 | 0.9709 | 0.9008 | TCD-DMOEA |
| | S2 | 0.69461 | 0.69353 | 0.86000 | 0.916407 | 0.9762 | **0.9891** | KF-DMOEA |
| | S3 | 0.72455 | 0.72170 | 0.9438 | 0.96196 | **0.99893** | 0.9826 | Tr-DMOEA |
| DF12 | S1 | **0.49625** | 0.47694 | 0.00105 | 0.06489 | 0.0045 | 0.3006 | DNSGA-II-A |
| | S2 | **0.53252** | 0.52852 | 0.00212 | 0.00185 | 0.00089 | 0.106 | DNSGA-II-A |
| | S3 | 0.49016 | 0.44808 | **0.60066** | 0.00174 | 0.00568 | 0.1009 | TCD-DMOEA |
| DF13 | S1 | 0.99324 | 0.99242 | **0.99794** | 0.995636 | 0.995 | 0.9087 | TCD-DMOEA |
| | S2 | 0.99485 | 0.99080 | 0.99706 | **0.99817** | 0.99721 | 0.9479 | PPS |
| | S3 | 0.99406 | 0.99434 | **0.99781** | 0.99708 | 0.99611 | 0.9361 | TCD-DMOEA |
| DF14 | S1 | 0.92624 | 0.78856 | **0.94998** | 0.92584 | 0.927 | 0.7855 | TCD-DMOEA |
| | S2 | 0.76854 | 0.75249 | **0.94998** | 0.82129 | 0.90391 | 0.826 | TCD-DMOEA |
| | S3 | 0.92649 | 0.81270 | **0.94996** | 0.93450 | 0.91622 | 0.772 | TCD-DMOEA |
| F5 | S1 | 0.38698 | 0.47343 | **0.94583** | 0.34478 | 0.6768 | 0.6133 | TCD-DMOEA |
| | S2 | 0.45555 | 0.48011 | **0.86738** | 0.45613 | 0.67023 | 0.7623 | TCD-DMOEA |
| | S3 | 0.57974 | 0.58742 | **0.94356** | 0.58405 | 0.59982 | 0.6696 | TCD-DMOEA |

**Table 3.** *Cont.*

| Functions | Settings | DNSGA-II-A | DNSGA-II-B | TCD-DMOEA | PPS | Tr-DMOEA | KF-DMOEA | Winner |
|---|---|---|---|---|---|---|---|---|
| F6 | S1 | 0.52946 | 0.50249 | **0.96014** | 0.46420 | 0.6917 | 0.5548 | TCD-DMOEA |
| | S2 | 0.61363 | 0.58633 | **0.96502** | 0.56726 | 0.67553 | 0.6745 | TCD-DMOEA |
| | S3 | 0.56928 | 0.59685 | **0.97848** | 0.66460 | 0.59692 | 0.4971 | TCD-DMOEA |
| F7 | S1 | 0.49083 | 0.50185 | **0.97899** | 0.558975 | 0.6482 | 0.5175 | TCD-DMOEA |
| | S2 | 0.53120 | 0.57648 | **0.97198** | 0.46678 | 0.64896 | 0.7871 | TCD-DMOEA |
| | S3 | 0.54482 | 0.60665 | **0.98071** | 0.73543 | 0.59342 | 0.5414 | TCD-DMOEA |
| F8 | S1 | 0.99865 | 0.99941 | 0.99998 | 0.99926 | **1** | 0.9871 | Tr-DMOEA |
| | S2 | 0.99977 | 0.99991 | 0.99998 | 0.99977 | **1** | 0.998 | Tr-DMOEA |
| | S3 | 0.99908 | 0.99993 | 0.99997 | 0.99997 | **1** | 0.9835 | Tr-DMOEA |
| F9 | S1 | 0.44287 | 0.33825 | **0.89925** | 0.29784 | 0.6831 | 0.5528 | TCD-DMOEA |
| | S2 | 0.43699 | 0.333972041 | **0.96078** | 0.26047 | 0.58884 | 0.6172 | TCD-DMOEA |
| | S3 | 0.52120 | 0.42007 | **0.90454** | 0.40561 | 0.64952 | 0.5602 | TCD-DMOEA |
| F10 | S1 | 0.65458 | 0.68298 | **0.98527** | 0.48178 | 0.6699 | 0.5614 | TCD-DMOEA |
| | S2 | 0.52382 | 0.46490 | **0.95732** | 0.50696 | 0.69461 | 0.7753 | TCD-DMOEA |
| | S3 | 0.70696 | 0.73979 | **0.99873** | 0.50912 | 0.77093 | 0.6982 | TCD-DMOEA |

**Table 4.** The average running time (seconds) of the six algorithms under the S2 configuration.

| Functions | DNSGA-II-A | DNSGA-II-B | TCD-DMOEA | PPS | Tr-DMOEA | KF-DMOEA |
|---|---|---|---|---|---|---|
| DF1 | 23.0258 | 27.0289 | 10.0028 | 30.1587 | 461.5591 | 30.4620 |
| DF2 | 29.8561 | 28.4196 | 9.8974 | 29.7459 | 363.2698 | 29.4512 |
| DF3 | 39.274 | 30.4785 | 10.0525 | 18.4148 | 459.2658 | 27.1253 |
| DF4 | 48.1478 | 30.0753 | 9.7598 | 17.4654 | 165.5802 | 23.3695 |
| DF5 | 29.4796 | 31.7592 | 8.8895 | 30.6544 | 399.0036 | 32.4890 |
| DF6 | 31.1548 | 29.2971 | 8.9536 | 24.8569 | 524.2016 | 21.0256 |
| DF7 | 43.4574 | 28.1490 | 9.2587 | 25.2584 | 378.0154 | 23.1016 |
| DF8 | 30.4765 | 29.0253 | 8.1258 | 28.5298 | 425.0545 | 27.4694 |
| DF9 | 30.4654 | 36.4695 | 8.5891 | 20.1489 | 484.0154 | 25.9726 |
| DF10 | 24.8989 | 24.6497 | 8.1856 | 89.2103 | 866.5962 | 97.6546 |
| DF11 | 21.4890 | 25.0245 | 9.9782 | 81.0365 | 890.4168 | 91.6469 |
| DF12 | 26.5982 | 21.4694 | 10.0159 | 49.2016 | 970.1460 | 47.4102 |
| DF13 | 19.0023 | 21.7592 | 7.9987 | 85.4160 | 1093.0001 | 98.7912 |
| DF14 | 18.8898 | 19.1654 | 12.0238 | 85.0489 | 908.4590 | 96.4694 |
| F5 | 132.0001 | 32.0795 | 13.0173 | 48.2203 | 513.4169 | 59.8591 |
| F6 | 63.8994 | 31.5911 | 13.5924 | 50.1605 | 545.2416 | 60.4991 |
| F7 | 54.6565 | 57.2259 | 12.3654 | 59.2056 | 571.1560 | 61.1697 |
| F8 | 32.7879 | 56.7411 | 13.2485 | 76.8569 | 1269.2036 | 86.7952 |
| F9 | 58.9891 | 39.2899 | 12.2498 | 52.5892 | 970.2596 | 63.7961 |
| F10 | 57.4590 | 29.0595 | 11.1475 | 49.0412 | 597.5269 | 52.5610 |

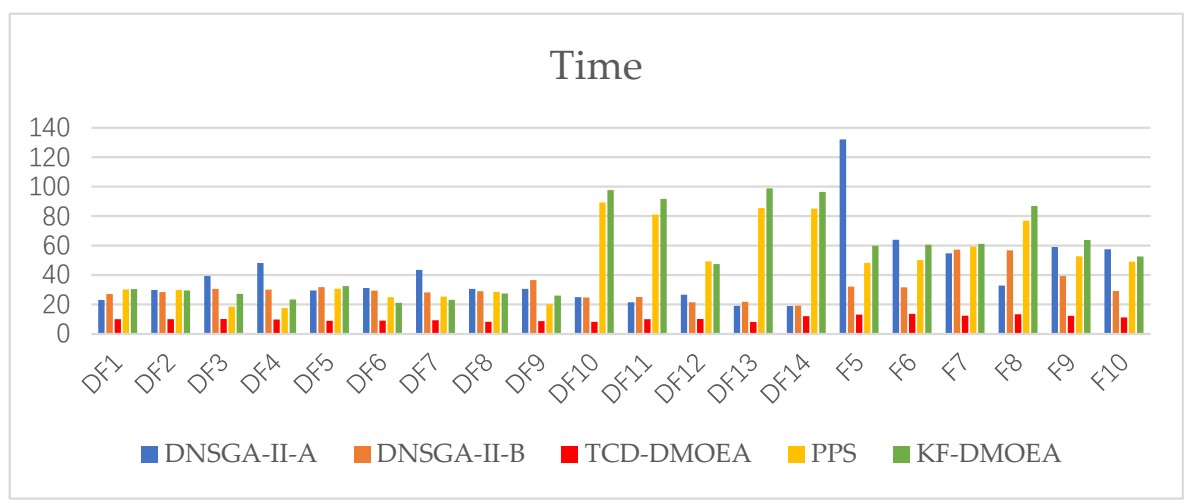

**Figure 6.** Average running time (s) obtained by comparing algorithms under the configuration of $n_t = 5$, $\tau_t = 10$.

## 5. Conclusions

In recent years, transfer learning has been proven to be one of the effective means to solve dynamic multi-objective optimization problems. However, the efficiency of transfer learning (also known as negative transfer) decreases significantly when the target domain is poorly similar to the source domain, or when the transfer learning method is incorrect. Negative transfer forces a search for solutions in the wrong direction, wasting a lot of computing resources. As a result, the running speed becomes slower and the convergence becomes worse.

In this article, a transfer learning method based on a cluster difference dynamic multi-objective optimization algorithm, TCD-DMOEA, was proposed. The TCD-DMOEA applies a clustering difference strategy to increase the similarity between the target domain and the source domain to reduce the likelihood of negative transfer, and the TradaBoost algorithm to classify good-quality populations. Therefore, when the environment changes drastically, the proposed method can improve the quality of the population in the drastically changing

environment, thereby improving the convergence and diversity of the dynamic multi-objective algorithm, and at the same time improve the running speed of the algorithm.

The above experimental results fully demonstrate that the algorithm significantly improves the performance of dynamic optimization. Compared with existing transfer learning-based algorithms, the proposed algorithms are tens or even hundreds of times faster at finding POS.

Although the proposed TCD-DMOEA can generate a high-quality initial population, the reliability of the acquired individuals becomes very poor. When the environmental changes are more complex, the accuracy of the cluster-based difference strategy will decrease, and due to the added classifier, the computational complexity of the algorithm will increase. Therefore, in future research work, we will explore the following promising directions. First, it would be beneficial for our static evolution process to try to combine multiple response mechanisms to cope with environmental changes, rather than employing a single strategy. Second, we can try to use classifiers with lower complexity to speed up the optimization process. Additionally, it will be worthwhile to test TCD-DMOEA on a wider range of problems with different types of variation.

**Author Contributions:** Conceptualization, F.Y.; methodology, F.Y.; software, F.Y.; validation, F.Y.; formal analysis, F.Y.; investigation, F.Y.; resources, F.Y.; data curation, F.Y.; writing—original draft preparation, F.Y.; writing—review and editing, F.Y. and G.-G.W.; visualization, F.Y.; supervision, G.-G.W. All authors have read and agreed to the published version of the manuscript.

**Funding:** This work was supported by National Key R&D Program of China (Grant No. 2022YFB3305300) and the Fundamental Research Funds for the Central Universities.

**Institutional Review Board Statement:** Not applicable.

**Informed Consent Statement:** Not applicable.

**Data Availability Statement:** Not applicable.

**Acknowledgments:** The authors are thankful to the anonymous reviewers for their valuable suggestions during the review process.

**Conflicts of Interest:** The authors declare no conflict of interest.

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
