# Peer review of "Transfer Learning Based on Clustering Difference for Dynamic Multi-Objective Optimization"

_applsci, doi:10.3390/app13084795_

Round 1

Reviewer 1 Report

1. Authors should quote equation numbers in text.

2. There are some typos like equation 1 after equation 16, table 2 is written twice in captions Etc. Please correct and check the complete paper.

3. There is no any winner column in table 3 as mentioned in text. Please correct.

4. Do not use ‘we’ in paper. Please check and correct it wherever it is written.

5. List of abbreviation is needed for better understanding.

6. Explain Figure 4.

7. How do authors use these algorithms in MATLAB. Please elaborate and show the MATLAB model for better understanding.

Reviewer 2 Report

Please find the attached report with comments

Round 2

Reviewer 1 Report

Comments/suggestions are addressed carefully.

Reviewer 2 Report

The authors improved their manuscript